# Asymptotic Behavior of the Coordinate Ascent Variational Inference in Singular Models

Sean Plummer[1], Anirban Bhattacharya[2], Debdeep Pati[3], Yun Yang[4]
[1]University of Arkansas, [2]Texas A&M University,
[3]University of Wisconsin-Madison, [4]University of Maryland
seanp@uark.edu, anirbanb@stat.tamu.edu, dpati2@wisc.edu, yy84@umd.edu

Mean-field approximations are widely used for efficiently approximating high-dimensional integrals. While the efficacy of such approximations is well understood for well-behaved likelihoods, it is not clear how accurately it can approximate the marginal likelihood associated with a highly non log-concave singular model. In this article, we provide a case study of the convergence behavior of coordinate ascent variational inference (CAVI) for a general $d$-dimensional singular model in standard form. We prove that such a model with real log canonical threshold (RLCT) $\lambda$ and multiplicity $m$, the CAVI system converges to one of $m$ locally attracting fixed points. Furthermore, at each of these fixed points, the evidence lower bound (ELBO) recovers the leading-order behavior of the asymptotic expansion of the log marginal likelihood predicted by Watanabe [1, 2, 3]. Our empirical results demonstrate that for models with multiplicity $m = 1$ the ELBO provides a tighter approximation to the log-marginal likelihood than Watanabe's [1] asymptotic approximation $-\lambda \log n + o(\log \log n)$ for a range of sample sizes.

## 1. Introduction

Variational inference (VI) [4–6] is an optimization-based approach to approximate Bayesian inference which aims to approximate the posterior distribution $p(w \mid X^n) \propto p(X^n \mid w)\pi(w)$ of the parameters of interest $w \in \mathcal{W}$ by selecting the best approximation $\hat{q}$ to the posterior within a pre-specified family of distributions $\mathcal{Q}$ by minimizing the Kullback-Leibler (KL) divergence

$$\hat{q}(w) = \operatorname*{argmin}_{q \in \mathcal{Q}} \int_{\mathcal{W}} q(w) \log \left[ \frac{q(w)}{p(w \mid X^n)} \right] dw.$$

This variational minimization problem is equivalent to maximizing a variational objective known as the evidence lower bound (ELBO)

$$\hat{q}(w) = \operatorname*{argmax}_{q \in \mathcal{Q}} \mathbb{E}_q \log \left[ \frac{p(X^n, w)}{q(w)} \right] := \operatorname*{argmax}_{q \in \mathcal{Q}} \operatorname{ELBO}(q),$$

which derives its name from the fact that it serves as a lower bound to the log marginal likelihood of the model; $\operatorname{ELBO}(q) \leq \log p(X^n)$. A well-known choice of variational family is the *mean-field variational family* which is comprised of product distributions over the parameters,

$$\mathcal{Q} := \{q(w) = \otimes_{i=1}^d q_i(w_i) \: : \: q \ll \pi \text{ a prob. measure on } \mathcal{W}\}.$$

The optimal variational approximation for the mean-field family is given by the product of the optimal marginal distributions $\hat{q}(w) := \otimes_j \hat{q}_j(w_j)$. Numerically, this distribution is computed using an algorithm known as coordinate ascent variational inference (CAVI); see CH 10 of Bishop [7] or the survey of Blei et al. [4] for further details on CAVI.

Although there is now a large body of work delineating the general statistical properties of mean-field variational inference [8–16], general results regarding the convergence properties of mean-field variational inference are limited in the literature. The algorithmic behavior of CAVI has been studied for several classes of models utilizing tools from dynamical systems with mixed results

Second Conference on Parsimony and Learning (CPAL 2025).

[17–21]. This approach works directly on the space of parameters $\mathcal{W}$ of the model and is forced to confront challenging technical issues in the form of intractable update equations and unwieldy normalization constants. More recent approaches have been able to provide general results for the mean-field approximation by considering the problem in the space of probability measures over $\mathcal{W}$. Lacker et al. [22] and Arnese and Lacker [23] provide general convergence results for the mean-field approximation and CAVI in the case of log-concave measures. Bhattacharya et al. [24] provide general convergence guarantees for two-block CAVI using techniques from convex optimization on Hilbert spaces.

In this work, we provide a case study of the convergence behavior of a block CAVI algorithm by considering the mean-field approximation of a $d$-dimensional *singular model* in standard form.[1] The standard form of a singular model arises from a deep result in algebraic geometry known as the resolution of singularities which shows that every singularity can be transformed into a simple normal crossing singularity [25, 26]. This transformation induces a new coordinate system which puts the posterior into its standard form; for additional details see Section 2.[2] The resolved coordinates are (typically) going to be made of blocks of the original coordinates $\mathcal{W}$, so the mean-field approximation to the standard form (in the resolved coordinates) is a block mean-field approximation of the original model (in the $\mathcal{W}$ coordinates). Furthermore, the standard form is highly nonconvex and does not satisfy standard relaxations of convexity such as the Polyak-Lojasiewicz (PL) condition [29]. Hence, the standard form provides an interesting case for which the previous results on the convergence of CAVI are not applicable.

We return to the dynamical systems approach to study the convergence properties of CAVI for a $d$-dimensional singular model in standard form in the asymptotic regime. We show that for a sufficiently large sample size $n$, CAVI converges to one of $m$ locally attracting fixed points, where $m$ is the *multiplicity* of the model; see Lemma 3.1 and Section 3.1.3 for more details. Furthermore, at each of these fixed points, the ELBO asymptotically recovers the correct leading-order asymptotic behavior of the log-marginal likelihood as predicted by Bhattacharya et al. [30]. Our theoretical guarantees are supported by our simulation study for a standard form of dimension $d = 4$. We empirically compare the ELBO and Watanabe's asymptotic expansion of the log marginal likelihood (logML) of the model [1–3]. Surprisingly, we find that there is a wide regime of sample size $n$ for which the ELBO provides a better approximation to the logML than the asymptotic approximation. The size of this regime depends on the dimension $d$ of the system. Finally, we provide a simple example that demonstrates the difference between the mean-field approximation in the original coordinate system and the mean-field approximation in the standard form coordinates.

Previous works on mean-field VI focus on specific examples of singular models [31–40] with the aim of bounding the real log canonical threshold (RLCT) to study the asymptotic behavior of the mean-field ELBO with respect to the original coordinate system. These asymptotic expansions recover similar leading-order behavior as Watanabe's asymptotic expansion, but the constants for the leading-order terms in the upper and lower bounds do not (typically) match. Although these approaches leverage dynamical systems techniques, they only work in the original coordinate system $\mathcal{W}$ of the model; due to the inability to compute the resolution map, i.e. the standard form coordinates, for general models.

The remainder of the article is organized as follows. In Section 2, we specify the problem setup and provide a brief review of some key concepts of singular model theory. Section 3 contains our main results on the convergence of the CAVI algorithm. Section 4 provides a simulation study on the convergence of the CAVI algorithm, which numerically corroborates our theoretical results. We further verify these results in the context of a simple singular regression model for which the

---

[1]The data generating distribution $q(x)$ is said to be *singular* for the model $\{p(x \mid w) \mid w \in \mathcal{W}\}$ if either the minimum locus of average log loss function $L(w) := -\mathbb{E}_X[\log p(x \mid w)]$ contains more than one point or there exists points in the minimum locus for which the Hessian matrix $\nabla^2 L$ fails to be positive definite.

[2]This coordinate system also exists for regular models [27, 28] and should not be confused with asymptotic normality.

resolution is known. Finally, we conclude our article and discuss related open problems in Section 5.

## 2. Background

### 2.1. Problem Setup and Notation

Let $X^n = (X_1, \ldots, X_n)^{\mathrm{T}}$ denote $n$ independent and identically distributed (i.i.d.) observations from a probability density function $p(x \mid w^\star)$. A Bayesian approach posits: (1) A statistical model $\{p(\cdot \mid w) : w \in \mathcal{W}\}$, where $\mathcal{W} \subset \mathbb{R}^d$ is compact, and (2) a prior (probability) distribution $\varphi(\cdot)$ on $\mathcal{W}$. The posterior distribution is given by $p(w \mid X^n) = e^{-nL_n(w)}\varphi(w)/p(X^n)$ with $L_n(w) := -\frac{1}{n}\sum_{i=1}^n \log p(X_i \mid w)$ the negative average log-likelihood function and $p(X^n) = \int_W e^{-nL_n(w)}\varphi(w)dw$ is the marginal likelihood (or evidence). We denote the average log loss function by $L(w) = -\mathbb{E}_X[\log p(x \mid w)]$, the empirical log-likelihood ratio by $K_n(w) = L_n(w^\star) - L_n(w)$, and the average log-likelihood ratio by $K(w) = L(w^\star) - L(w)$. We will denote the ELBO of a probability density $\rho$ with respect to another probability density $\mu$ by $\mathrm{ELBO}_\mu(\rho) := E_\rho[\log \mu] - E_\rho[\log \rho]$.

### 2.2. Singular Models

Watanabe [1, 2, 3] shows for any singular model satisfying some mild technical conditions, the asymptotic behavior of the log-marginal likelihood follows

$$L_n(w^\star) - \lambda \log n + (m-1)\log(\log n) + R_n, \tag{2.1}$$

assuming the data is generated from $P^\star \equiv p(\cdot \mid w^\star)$, with the stochastic error term $R_n = O_{P^\star}(1)$. The quantity $\lambda \in (0, d/2]$ is called the *real log-canonical threshold* (RLCT) and serves as a generalized measure of dimension for the parameter space of the model. The integer $m \geq 1$ is the *multiplicity* and measures how many parameters achieve this dimension. Unlike the asymptotic expansion for regular models, the RLCT and its multiplicity depend on both the model and the true data generating distribution. For a regular statistical model, we have $(\lambda, m) = (d/2, 1)$ and the expansion (2.1) reduces to the usual Laplace approximation. A pivotal part of the derivation of this asymptotic expansion is a deep mathematical result in algebraic geometry known as the resolution of singularities [25, 26]. This result guarantees a family of local coordinates $\{U_\ell\}$ and transformations $\{w = g_\ell(u)\}$, here-in referred to as the *resolved coordinate system*, which allows us to view the singular model as a regular model in a higher dimensional space. In each part of this resolved coordinate system $(U_\ell, g_\ell)$, the normalized posterior of the model is expressed in its *standard form*,

$$p(g_\ell(u) \mid X^n) \propto u^{\mathrm{h}_\ell} \exp\{-nu^{2\mathrm{k}_\ell}\}b_\ell(u), \ u \in [0,1]^d,$$

where $u^{2\mathrm{k}_\ell} := u_1^{2k_{\ell,1}}u_2^{2k_{\ell,2}}\cdots u_d^{2k_{\ell,d}}$ is a monomial with multi-index $\mathrm{k}_\ell = (k_{\ell,1}, \ldots, k_{\ell,d})^{\mathrm{T}} \in \mathbb{N}^d$ having at least one positive entry, $u^{\mathrm{h}_\ell} := u_1^{h_{\ell,1}}u_2^{h_{\ell,2}}\cdots u_d^{h_{\ell,d}}$ a monomial with multi-index $\mathrm{h}_\ell = (h_{\ell,1}, \ldots, h_{\ell,d})^{\mathrm{T}} \in \mathbb{N}^d$, and $b_\ell(u) > 0$ is a real-analytic function. In each local coordinate system $U_\ell$, the multi-indexes $\mathrm{k}_\ell$ and $\mathrm{h}_\ell$ of (local) standard form of a model can be used to determine the local RLCT $\lambda_\ell$ and the multiplicity $m_\ell$ via simple closed-form expressions, with $\lambda_\ell = \min_{j \in [d]}(h_{\ell,j} + 1)/(2k_{\ell,j})$ and $m_\ell = \#\{i \in [d] : (h_{\ell,i} + 1)/(2k_{\ell,i}) = \lambda_\ell\}$. The (global) RLCT $\lambda$ and multiplicity $m$ of the singular model are defined by $\lambda = \min_\ell \lambda_\ell$, $m = \max\{m_\ell : \lambda_\ell = \lambda\}$.

Although the resolution coordinates $\{g_\ell\}$ are theoretically guaranteed to exist, computing them is not feasible for most statistical models [27, 28]. Beyond simple examples and the reduced rank regression model [35] the determination of the RLCT and multiplicity remains a challenging open problem. Several works have made progress in this direction by providing upper bounds for the RLCT in several classes of singular models [41–47]. For textbook level treatments of singular models see Watanabe [27] or Watanabe [28]. For recent survey and an alternative derivation of Equation (2.1) using only probabilistic tools see Bhattacharya et al. [30]. For more on model selection in singular settings, we refer the reader to Watanabe [48], Drton and Plummer [49], or Wang and Yang [40].

# 3. Convergence of CAVI Algorithm

Our main result shows that for any singular model of parameter dimension $d$, the optimal mean-field approximation to the standard form of the model computed using the CAVI algorithm converges to a local fixed point that recovers the correct leading-order behavior [30]. Our analysis reveals that both the number of possible fixed points and the rate of convergence of the system depend on the multiplicity of the model. The proof is based on the dual view of the CAVI algorithm as a discrete time dynamical system of the variational parameters. A brief introduction to dynamical systems can be found in Appendix A of Plummer et al. [21]. For textbook-level treatments, see Wiggins [50], Kuznetsov [51], or Elaydi [52].

## 3.1. CAVI for Standard Form

We begin by defining notation for several functions which will play a pivotal role in the study of the dynamics of the CAVI system associated to the singular model in standard form. Define the density function $f_{k,h,\beta}(u) = u^h \exp(-\beta u^{2k})/B(k,h,\beta)$, $u \in [0,1]$, where $B(k,h,\beta) = \int_0^1 x^h \exp(-\beta x^{2k})\,dx$. The quantity $G(\lambda,\beta) := \int_0^1 u^{2k} f_{k,h,\beta}(u)\,du$ depends on $k$ and $h$ only through $\lambda = (h+1)/(2k)$. A straightforward integration by parts shows that $G(\lambda,\beta) = [\lambda\gamma(\lambda + 1,\beta)]/[\beta\gamma(\lambda,\beta)]$, where for any $a > 0$, $\gamma(a,x) = \int_0^x t^{a-1}e^{-t}dt/\Gamma(a)$, $x > 0$, is the cumulative distribution function (CDF) of the Gamma distribution with shape parameter $a$ and rate parameter 1.

Consider the following standard form of the singular model with dimension $d \geq 2$,

$$\gamma_K(u_1,\ldots,u_d) \propto u^{\mathrm{h}} \exp\{-nu^{2\mathrm{k}}\}, \quad u \in [0,1]^d,$$

where $u^{2\mathrm{k}} = \prod_{j=1}^d u_j^{2k_j}$, $u^{\mathrm{h}} = \prod_{j=1}^d u_j^{h_j}$, and $b(u) \equiv 1$ for simplicity.[3] The optimal marginals for the mean-field approximation to the standard form of the posterior $\gamma_K(\mathbf{u})$ are computed using the CAVI algorithm [4]. For each $j \in [d] := \{1,2,\ldots,d\}$, the $j$-th marginal distribution at the $(t+1)$-th iteration of the CAVI algorithm for a model in standard form is computed by

$$\rho_j^{(t+1)}(u_j) \propto \exp\left\{\int_{[0,1]^{d-1}} \log[\gamma_K(\mathbf{u})]\rho_{-j}^{(t+1)}(\mathbf{u}_{-j})d\mathbf{u}_{-j}\right\},$$

where $\rho_{-j}^{(t+1)}(\mathbf{u}_{-j}) = \rho_1^{(t+1)}(u_1) \otimes \cdots \otimes \rho_{j-1}^{(t+1)}(u_{j-1}) \otimes \rho_{j+1}^{(t)}(u_{j+1}) \otimes \cdots \otimes \rho_d^{(t)}(u_d)$. A straightforward computation shows that the $j$-th marginal distribution is given by, $\rho_j^{(t+1)}(u_j) = f_{k_j,h_j,n\mu_j^{(t+1)}}(u_j)$, $j \in [d]$, where for $t \geq 0$ and $j \in [d]$,

$$\mu_j^{(t+1)} = \prod_{s=1}^{j-1} G(\lambda_s, n\mu_s^{(t+1)}) \cdot \prod_{s=j+1}^{d} G(\lambda_s, n\mu_s^{(t)}). \tag{3.1}$$

The ELBO at the $(t+1)$-th iteration of the algorithm, which is computed using $\rho^{(t+1)}(\mathbf{u}) = \otimes_j \rho_j^{(t+1)}(u_j)$, is given by,

$$\mathrm{ELBO}_{\gamma_K}(\rho^{(t+1)}) = -n\prod_{s=1}^{d} G(\lambda_s, n\mu_s^{(t+1)}) + \sum_{s=1}^{d} n\mu_s^{(t+1)} G(\lambda_s, n\mu_s^{(t+1)}) + \sum_{s=1}^{d} \log B(k_s, h_s, n\mu_s^{(t+1)}).$$

Notice that both the optimal $j$-th marginal distribution $\rho_j^{(t+1)}(u_j)$ and the ELBO are fully determined by the behavior of the dynamical system in the variational parameters $\mu_j^{(t+1)}$ in Equation

---

[3]A key part of Watanabe's analysis is the study of $\mathscr{Z}_K(n) = \int_{[0,1]^d} \exp\{-nK(w)\}dw$, where $K(w)$ is average log-likelihood ratio function. The notation for the deterministic standard form $\gamma_K(u) = \exp\{-nK(g(u))\}|g'(u)|/\mathscr{Z}_K(n) = u^{\mathrm{h}} \exp\{-nu^{2\mathrm{k}}\}b(u)/\mathscr{Z}_K(n)$, where $g(u) = w$ is the resolution map, arises from this.

3.1. Convergence of the CAVI algorithm translates to the convergence of the variational dynamical system to a fixed point. A fixed point $\mu^*$ of the CAVI dynamical system must statisfy,

$$\mu_j^* = \prod_{s \neq j} G(\lambda_s, n\mu_s^*), \text{ for all } 1 \leq j \leq d.$$

At such a fixed point, ELBO is given by,

$$\text{ELBO}_{\gamma_K}(\rho^*) = -n \prod_{s=1}^{d} G(\lambda_s, n\mu_s^*) + \sum_{s=1}^{d} n\mu_s^* G(\lambda_s, n\mu_s^*) + \sum_{s=1}^{d} \log B(k_s, h_s, n\mu_s^*).$$

Hence, understanding the number and nature of fixed points of the CAVI dynamical system (3.1) determines the behavior of the ELBO.

We now state our main theorem, which characterizes these fixed points and shows that each yields an ELBO asymptotically matching $-\lambda \log n + C$.

**Theorem 3.1.** *For a standard form with RLCT $\lambda$ and multiplicity $m$, the recursive formula (3.1) admits $m$ locally attracting fixed points for large sample size $n$. At each of these fixed points, the ELBO admits the asymptotic expansion $\text{ELBO}_{\gamma_K}(\rho^*) \asymp -\lambda \log n + C$.*

The study of the convergence behavior of a dynamical system is called a *local stability analysis* of the system and proceeds in two steps. First, we determine the fixed points of the system in Lemma 3.1. We will show that the number of fixed points is related to the multiplicity $m$ of the singular model. Next, Lemma A.2 provides the partial derivatives for the CAVI system. We show that for a sufficiently large sample size $n$, each of these $m$ fixed points is *locally attracting* by showing that the spectral radius of the Jacobian of the system at the fixed point is strictly less than 1. This guarantees that the system will converge to the fixed point if it is initialized sufficiently close to that fixed point. At each of these fixed points, applying the asymptotic computations from Bhattacharya et al. [30] shows that the ELBO corresponding to this fixed point is of order $-\lambda \log n + C$.

### 3.1.1. Fixed Points of CAVI

We begin by computing the fixed points of the system. The number of fixed points depend on the model's multiplicity $m$. For a system with multiplicity $m$, the RLCTs satisfy

$$\lambda = \lambda_1 = \cdots = \lambda_m < \lambda_{m+1} \leq \cdots \leq \lambda_d.$$

The fixed points of this system are given by the following lemma, the proof of which can be found in Appendix A.1,

**Lemma 3.1.** *For a standard form with RLCT $\lambda$ and multiplicity $m$, the recursive system (3.1) admits a fixed point $\mu^* = (\mu_1^*, \ldots, \mu_d^*)^T \in (0,1)^d$ given by*

$$\mu_j^* = C_j n^{\frac{1-m}{m}}, \quad 1 \leq j \leq m, \quad \mu_s^* = C_s n^{-1}, \quad m+1 \leq s \leq d,$$

*where the constants satisfy the system of equations*

$$\prod_{j=1}^{m} C_j = \lambda^{m-1} \prod_{m+1 \leq s \leq d} G(\lambda_s, C_s), \quad C_s G(\lambda_s, C_s) = \lambda, \quad m+1 \leq s \leq d.$$

*Furthermore, for a sufficiently large sample size $n$, the number of fixed points in the system is equal to its multiplicity $m$.*

### 3.1.2. Local Stability of Fixed Point

Next, we need to determine the spectral radius of the Jacobian at these fixed points. To do this, we need to compute the Jacobian of the CAVI system. In order to shorten the notation for the partial derivatives for the CAVI equations $\partial \mu_j^{(t+1)} / \partial \mu_k^{(t)}$ we introduce the function

$$R(\lambda, \beta) := \frac{\beta^{\lambda+1} \exp(-\beta)}{\Gamma(\lambda+1)\gamma(\lambda+1, \beta)} - \frac{\beta^{\lambda} \exp(-\beta)}{\Gamma(\lambda)\gamma(\lambda, \beta)}.$$

The derivative of $G$ with respect to $\beta$ is given by

$$\frac{d}{d\beta}G(\lambda,\beta) = \frac{G(\lambda,\beta)}{\beta}(-1 + R(\lambda,\beta)).$$

A tedious series of recursive computations, found in Appendix A.2, yields the Jacobian of the CAVI system.

**Lemma 3.2.** *The partial derivatives of the recursive system* (3.1) *are given by*

$$\frac{\partial\mu_k^{(t+1)}}{\partial\mu_1^{(t)}} = 0, \quad 1 \le k \le d, \quad \frac{\partial\mu_1^{(t+1)}}{\partial\mu_k^{(t)}} = \frac{\mu_1^{(t+1)}}{\mu_k^{(t)}}(-1 + R(\lambda_k, n\mu_k^{(t)})), \ 2 \le k \le d,$$

$$\frac{\partial\mu_j^{(t+1)}}{\partial\mu_k^{(t)}} = \frac{\mu_j^{(t+1)}}{\mu_k^{(t)}} \prod_{k \le p < j} R(\lambda_p, n\mu_p^{(t)})(-1 + \prod_{1 \le \ell < k} R(\lambda_\ell, n\mu_\ell^{(t+1)}))(-1 + R(\lambda_k, n\mu_k^{(t)})), \quad 2 \le k < j \le d,$$

$$\frac{\partial\mu_k^{(t+1)}}{\partial\mu_k^{(t)}} = (-1 + R(\lambda_k, n\mu_k^{(t+1)}))(-1 + \prod_{1 \le \ell < k} R(\lambda_\ell, n\mu_\ell^{(t+1)})), \quad 2 \le k \le d,$$

$$\frac{\partial\mu_j^{(t+1)}}{\partial\mu_k^{(t)}} = \frac{\mu_j^{(t+1)}}{\mu_k^{(t+1)}} \prod_{1 \le \ell < j} R(\lambda_\ell, n\mu_\ell^{(t+1)})(-1 + R(\lambda_k, n\mu_k^{(t+1)})), \quad 2 \le j < k \le d.$$

At this fixed point $R(\lambda, n\mu_j^*)$ is exponentially small in $n$ for each $1 \le j \le m$. It follows from the Gershgorin circle theorem [53, 54], that the spectral radius of the Jacobian at this fixed point is strictly less than 1. This shows that the fixed point $\mu^*$ is a hyperbolic attracting fixed point of the system. Initializing the system sufficiently close to this fixed point guarantees that the system will converge to this fixed point [50–52].

Combining these two lemmas together with the bounds from Bhattacharya et al. [30] shows that for any $d$-dimensional singular model, numerically computing the mean-field approximation to the posterior in standard form, i.e. in the coordinate system of the resolution mapping, always produces an ELBO which recovers the correct leading-order behavior of the log-marginal likelihood.

### 3.1.3. Consequences of Multiplicity

The multiplicity of the singular model plays a rather surprising role in the convergence properties of the CAVI system. First, the multiplicity $m$ and the sample size $n$ determine the number of fixed points in the system. For a small to moderate sample size $n$, the system has a single fixed point of attraction for which $C_1 = C_2 = \cdots = C_m$. For a large sample size $n$, the system undergoes a *bifurcation* — a change in the behavior of the dynamical system as one of the parameters changes — in which the original fixed point becomes asymptotically repelling and the system develops $m$ locally attracting fixed points. Second, the multiplicity of the system also determines the convergence speed of the CAVI algorithm to a fixed point. Specifically, systems with larger multiplicity converge at a slower rate than systems with smaller multiplicity. To simplify the discussion, consider the case where $n$ is sufficiently large. The eigenvalues of the CAVI system Jacobian are exponentially close to $1 - R(\lambda_j, n\mu_j^*)$, for $2 \le j \le d$. For $2 \le j \le m$, $R(\lambda, n\mu_j^*) = O(\exp\{-Cn^{1/m}\})$ hence, each eigenvalue is exponentially close to 1, and at each time step, the dynamical system moves very little in these directions.

## 4. Simulation Study for CAVI in Standard Forms

We now present numerical experiments for the standard form $\gamma_K(u) = u^h \exp\{-nu^{2k}\}/\mathcal{Z}_K(n)$, illustrating Theorem 3.1 and the behavior of multiplicity. We also compare the ELBO to Watanabe's asymptotic formula (2.1), finding that for a broad regime of $n$, the ELBO can be more accurate. Finally, we give a simple singular regression example in which the ELBO of the mean-field approximation in the original parameter coordinates fails to recover the correct leading-order term $-\lambda \log n$, while the ELBO of the mean-field approximation in the resolved standard-form coordinates do indeed recover it.

| $(\lambda_1, \lambda_2, \lambda_3, \lambda_4)$ | Parameter | Dynamic | Numeric | Product |
|---|---|---|---|---|
| Case 1: $(1/4, 1/3, 1/3, 1/2)$ | $C_1$ | 0.005410077 | | 0.005410609 |
| | $C_2$ | 1.7167214 | 1.716581 | |
| | $C_3$ | 1.7167214 | 1.716581 | |
| | $C_4$ | 0.9799794 | 0.979926 | |
| Case 2: $(1/3, 1/3, 1/2, 1/2)$ | $C_1$ | 0.62576484 | | 0.01620921 |
| | $C_2$ | 0.02590196 | | |
| | $C_3$ | 1.511632 | 1.511556 | |
| | $C_4$ | 1.511632 | 1.511556 | |
| Case 3: $(1/3, 1/3, 1/3, 1/2)$ | $C_1$ | 0.4631405 | | 0.02450186 |
| | $C_2$ | 0.2348403 | | |
| | $C_3$ | 0.2252707 | | |
| | $C_4$ | 1.511632 | 1.511556 | |
| Case 4: $(1/5, 1/5, 1/5, 1/5)$ | $C_1$ | 0.2989035 | | 0.008 |
| | $C_2$ | 0.2989076 | | |
| | $C_3$ | 0.2989061 | | |
| | $C_4$ | 0.2989020 | | |

Table 1: A table containing the coefficients for the fixed points of the mean-field equations for various combination of $\lambda$ for $n = \lfloor \exp(13) \rfloor$. The *Dynamic* column records the values of $C_j$ , for $1 \leq j \leq d$, computed using the CAVI fixed points. The *Numeric* column contains the numerically approximate solution to the analytic equation $C_s G(\lambda_s, C_s) = \lambda$, for $m + 1 \leq s \leq d$. The *Product* column contains the value of the formula $\lambda^{m-1} \prod_{m+1 \leq s \leq d} G(\lambda_s, C_s)$ computed using the values from the *Numeric* column. In each case, the coefficients $C_s$, $m + 1 \leq s \leq d$, in Dynamics column approximately match the corresponding quantity in the Numerical column. The product of the Dynamic column coefficients $C_j$, $1 \leq j \leq m$, approximately equal the value in the Product column.

## 4.1. CAVI in the Standard Form

We consider a $d = 4$ dimensional standard form $\gamma_K(u) = u^{\mathrm{h}} \exp\{-nu^{2\mathrm{k}}\}/\mathcal{Z}_K(n)$ with four different choices of multi-indices $(\mathrm{k}, \mathrm{h})$.[4] Concretely, we study:

1. $\mathrm{k} = (2, 3, 3, 1)$ and $\mathrm{h} = (1, 2, 2, 0)$ with $(\lambda, m) = (1/4, 1)$.
2. $\mathrm{k} = (3, 3, 2, 1)$ and $\mathrm{h} = (1, 1, 0, 0)$ with $(\lambda, m) = (1/3, 2)$.
3. $\mathrm{k} = (3, 3, 3, 1)$ and $\mathrm{h} = (1, 1, 1, 0)$ with $(\lambda, m) = (1/3, 3)$.
4. $\mathrm{k} = (5, 5, 5, 5)$ and $\mathrm{h} = (1, 1, 1, 1)$ with $(\lambda, m) = (1/5, 4)$.

For each sample size $n$, we initialize CAVI, run to a tight convergence tolerance of $10^{-12}$, and record the converged variational parameters $\mu^*$ and $\mathrm{ELBO}_{\gamma_K}(\rho^*)$.

### 4.1.1. Validating Lemma 3.1

Table 1 shows, for each case at $n = \lfloor e^{13} \rfloor$ initialized from $\mu^{(0)} = 0$, the coefficients $\{C_j\}$ from the fixed point $\mu_j^*$, for $1 \leq j \leq 4$. The column labeled "Dynamic" shows $C_j$ as obtained via the CAVI system, while the column labeled "Numeric" and "Product" display a direct numerical solution of the consistency conditions $C_s G(\lambda, C_s) = \lambda$ and $\lambda^{m-1} \prod_{m+1 \leq s \leq d} G(\lambda_s, C_s)$, respectfully. The close match across columns verifies that the fixed-point equations in Lemma 3.1 hold.

### 4.1.2. Validating Theorem 3.1

Next, we check whether $\mathrm{ELBO}_{\gamma_K}(\rho^*)$ matches $-\lambda \log n + C$ for large $n$. Table 2 displays the least-squares estimates $\hat{\beta}_0, \hat{\beta}_1, \hat{\beta}_2$ from regressing $\mathrm{ELBO}_{\gamma_K}(\rho^*)$ onto $\beta_0 + \beta_1 \log n + \beta_2 \log \log n$. For each

---

[4]The resolution map does not need to be computed in this case as the standard form is the posterior of the model expressed in the resolution coordinates.

| Case | $(\lambda, m)$ | $\beta_0$ Estimate (P-val) | $\beta_1$ Estimate (P-val) | $\beta_2$ Estimate (P-val) |
|------|---------------|---------------------------|---------------------------|---------------------------|
| Case 1 | $(1/4, 1)$ | 2.513e+00 (2e-16) | $-2.500$e-01 (2e-16) | 5.879e-13 (0.816) |
| Case 2 | $(1/3, 2)$ | 2.133e+00 (2e-16) | $-3.333$e-01 (2e-16) | 2.729e-10 (0.285) |
| Case 3 | $(1/3, 3)$ | 2.700e+00 (2e-16) | $-3.333$e-01 (2e-16) | 9.589e-09 (0.027) |
| Case 4 | $(1/5, 4)$ | 4.8774074 (2e-16) | $-0.2003880$ (2e-16) | 0.0065515 (0.000541) |

Table 2: A table containing the estimated coefficients and P-values corresponding to the regressions $\text{ELBO}(q^*) = \beta_0 + \beta_1 \log n + \beta_2 \log \log n$ when ELBO is computed using mean-field VI. As predicted by Theorem 3.1, we see that the $\beta_1$ estimate is approximately $-\lambda$ and the ELBO fails to correctly recover the lower order $\log \log n$ term associated with the multiplicity $m$ when $m > 1$ as the $\beta_2$ estimate is approximately $0$ instead of the correct $(m-1)$ value predicted by Watanabe's asymptotic expansion.

case, if we were to fit a regression of the form $\beta_0 + \beta_1 \log n + \beta_2 \log \log n$ to the log-marginal likelihood $\log \mathcal{Z}_K(n)$, then the estimated regression coefficients should be approximately $\hat{\beta}_1 \approx -\lambda$ and $\hat{\beta}_2 \approx (m-1)$ based on Watanabe's asymptotic expansion [28]. In contrast, we see that the estimated coefficient are $\hat{\beta}_1 \approx -\lambda$ and $\hat{\beta}_2 \approx 0$. Hence, the ELBO fails to recover the correct $\log \log n$ term for models with multiplicity $m > 1$. This is exactly what we expect based on Theorem 3.1.

### 4.1.3. Comparing the ELBO and Asymptotic Expansions

We also compare the ELBO and the asymptotic expansion in 2.1 as approximations to the log marginal-likelihood (logML) for standard forms of various dimensions $d \in \{4, 8, 10, 20\}$, each with RLCT $\lambda = 1/2$ and multiplicity $m = 1$.[5] The multiplicity $m = 1$ is chosen so that the ELBO matches the true asymptotic expansion of the log-marginal likelihood and the bias, the difference between the ELBO and the log-ML, depends only on the constant terms. Additional simulations for lower-dimensional models are provided in Appendix A.4.

Surprisingly, our results in Figure 1 demonstrate that the ELBO provides a better approximation to the logML than the asymptotic approximation $-\lambda \log n$ in Equation (2.1) for a large "pre-asymptotic" regime of sample sizes $n$. Eventually, in the "asymptotic" regime, the asymptotic approximation $-\lambda \log n$ is better a approximation to the logML than the ELBO. This is due to the constant bias term that arises in the ELBO. In this context, bias is the difference between the ELBO and the log-ML, $|\text{ELBO}_{\gamma_K}(\rho^*) - \log \mathcal{Z}_K(n)|$. As $n \to \infty$, the approximation error for the ELBO converges to a constant that depends on the dimension of the model. Furthermore, these simulations suggest that the point at which the ELBO begins to diverge significantly from the logML also depends on the dimension of the model; note that the point at which the asymptotic approximation error dips below the ELBO approximation error moves right as $d$ increases from $d = 4$ to $d = 20$.

We now outline the dimensional dependence of the bias term. In order to simplify the following analysis, let us consider the case $\lambda = \lambda_1 < \lambda_2 \leq \lambda_3 \leq \cdots \leq \lambda_d$. The multiplicity of the system is $m = 1$ and the CAVI fixed point is $\mu^* = (C_1, C_2/n, C_3/n, \ldots, C_d/n)$, where $C_1 = \prod_{s=2}^d G(\lambda_s, C_s)$ and $C_s G(_s, C_s) = \lambda$. In this case the leading order term in the ELBO is $B(k_1, h_1, n\mu_1^*) \asymp -\lambda \log n$ which cancels out the leading order term of the asymptotic expansion of the log-ML. The bias becomes

$$-n \prod_{s=1}^d G(\lambda_s, n\mu_s^*) + \sum_{s=1}^d n\mu_s^* G(\lambda_s, n\mu_s^*) + \sum_{s=2}^d \log B(k_s, h_s, n\mu_s^*).$$

Plugging in $\mu^* = (C_1, C_2/n, C_3/n, \ldots, C_d/n)$ and reducing terms we see that the first term and second term in the sum reduce to $(d-1) \cdot \lambda$. The remaining term can be bound by

$$(d-1) \log B(k_2, h_2, C_2) \leq \sum_{s=2}^d \log B(k_s, h_s, C_s) \leq d-1) \log(d-1) \log B(k_d, h_d, C_d).$$

---
[5]This choice of dimension is due to computational limitations.

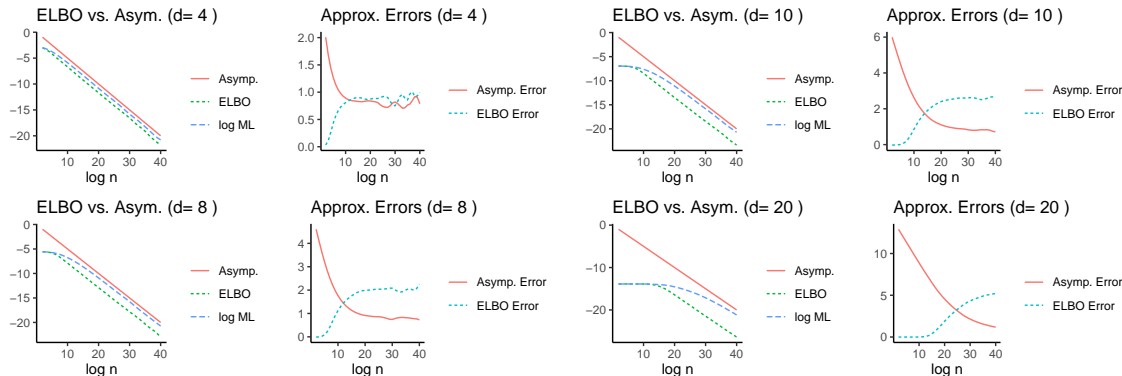

Figure 1: **ELBO vs Asymp.**: A plot of the log marginal likelihood (logML), the ELBO, and Watanabe's asymptotic expansion for a $d \in 4, 8, 10, 20$ dimensional standard form with RLCT $\lambda = 1/2$ and multiplicity $m = 1$. Surprisingly, in the non-asymptotic regime, the ELBO provides a better approximation of the log marginal likelihood than the asymptotic expansion. **Approx. Errors**: A plot of the approximation errors between the logML and ELBO (ELBO Error) and between the logML and asymptotic approximation (Asymp. Error). As $n \to \infty$, both errors converge to constant values.

which makes the overall bias on the order of $d - 1$

$$(d - 1) \cdot [\lambda + \log B(k_2, h_2, C_2)] \leq \text{Bias} \leq (d - 1) \cdot [\lambda + \log B(k_d, h_d, C_d)].$$

## 4.2. Singular Regression Example

Previous work on mean-field VI in singular models [31, 39] demonstrates that the leading coefficient of the asymptotic expansion of the ELBO is not the RLCT when the mean-field approximation is computed in the original coordinate system of the model. We will now provide a simple example of a singular regression model for which the asymptotic behavior of the ELBO for the mean-field approximation in the original coordinate system does not behave like $-\lambda \log n + C$, but the ELBO for the mean-field approximation in the resolved coordinate system asymptotically behaves like $-\lambda \log n + C$. Additional details for this example, including a derivation of the resolution map, can be found in Appendix A.3.

We will consider the following singular regression model, Example 46 from [28], for $x, y \in \mathbb{R}$, and parameters $w = (a, b, c) \in [-1, 1]^3$,

$$p(x, y \mid a, b, c) = \frac{1}{2\sqrt{2\pi}} \exp \left\{ -\frac{1}{2}(y - aS(bx) - cx)^2 \right\} I_{[-1,1]^3}(x), \tag{4.1}$$
$$\varphi(a, b, c) = 1/8,$$

where $S(x) = x^2 + x$. We will assume that the true data generating distribution is $q(x, y) = p(x, y \mid 0, 0, 0)$. The average log-density ratio for this setting is $K(a, b, c) = (ab + c)^2/2 + a^2 b^4/6$, which is singular at 0. The RLCT and multiplicity for this singular model are $\lambda = 3/4$ and $m = 1$.

A numerical investigation of the asymptotic behavior of the ELBO for this system in the original coordinate system of the model and in each of the resolution coordinates is summarized in Table 3. In the original $(a, b, c)$-coordinates of the model, the model is not in standard form and the CAVI solution does not recover $-\lambda \log n$. a regression of the ELBO on $\beta_0 + \beta_1 \log n + \beta_2 \log \log n$ yields $\hat{\beta}_1 \approx -1$ instead of the true RLCT $\lambda = 3/4$. In local resolved coordinate system, the model is in standard form and the CAVI solution recovers ELBO $\approx -\lambda \log n + C$. Numerically we find $\hat{\beta}_1 \approx -\lambda_j$ in each local coordinate system $U_j$.

| Coordinate System | $(\lambda, m)$ | $\beta_0$ Estimate (P-val) | $\beta_1$ Estimate (P-val) | $\beta_2$ Estimate (P-val) |
|---|---|---|---|---|
| Original Coord. | NA | 2.6741243 (2e-16) | $-1.0103421$ (2e-16) | 0.0792974 (2e-16) |
| Res. Coord. 1 | $(1, 1)$ | $-1.063e{+}00$ (2e-16) | $-1.000e{+}00$ (2e-16) | $-5.789e{-}05$ (0.93) |
| Res. Coord. 2 | $(1, 2)$ | $-0.8947223$ (2e-16) | $-1.0233375$ (2e-16) | 0.1858234 (2e-16) |
| Res. Coord. 3 | $(3/4, 1)$ | $-1.398e{+}00$ (2e-16) | $-7.500e{-}01$(2e-16) | $-6.569e{-}05$ (5.21e-06) |
| Res. Coord. 4 | $(3/4, 1)$ | $-1.525e{+}00$ (2e-16) | $-7.500e{-}01$ (2e-16) | $-9.563e{-}06$ (0.613) |

Table 3: A table containing the estimated coefficients and P-values corresponding to the regressions $\mathrm{ELBO}(q_j^*) = \beta_0 + \beta_1 \log n + \beta_2 \log \log n$ when ELBO is computed using mean-field VI for the example in Equation 4.1. The original coordinate system incorrectly recovers the largest of the local RLCTs instead of the smallest one.

## 5. Conclusion

In this article, we analyzed the dynamics of coordinate ascent variational inference (CAVI) for singular models in standard form, showing that:

1. There are $m$ stable fixed points when the model has multiplicity $m$.

2. In the "asymptotic" regime, each of these fixed points is asymptotically stable and the CAVI ELBO recovers the correct leading-order term, $-\lambda \log n$, of the log marginal likelihood.

Our experiments confirm these theoretical predictions and reveal that the ELBO can, in a "non-asymptotic" regime of sample sizes, yield better estimates of the log marginal likelihood than the classic asymptotic approximation. In the "asymptotic" regime, the classic asymptotic approximation provides a closer approximation of the log marginal likelihood than the ELBO. Interestingly we find that the point at which the system changes from the "non-asymptotic" regime to the "asymptotic" regime depends on the dimension of the system.

There are several open questions which are related to our work. First, in the example of Section 4.2, working in the original coordinates yields the wrong RLCT, but the correct RLCT emerges from the resolved coordinates. This suggests that while mean-field VI in the original coordinates may fail to produce the "true" leading term, it might still behave consistently for certain model-selection tasks; see [16] for recent results in singular mixture models. Second, the singular regression example also suggests that the resolution coordinates are needed for the mean-field approximation to recover the correct leading behavior of the log-marginal likelihood. Unfortunately, computation of the resolution coordinates is not feasible for all but the simplest singular models. It would be valuable to determine if transformation-based variational methods such as normalizing flows [55, 56] are able to by-pass the need for the computation of the resolution as part of the variational approximation. Finally, it would be of interest to determine if more flexible variational families such as black-box VI [57] or semi-implicit VI [58–60] would be able to provably recover the lower-order $\log \log n$ term in the asymptotic expansion of the ELBO.

## Acknowledgments

The authors would like to thank the anonymous referees for their constructive comments that improved the quality of this paper. AB acknowledges NSF DMS 2210689, NSF DMS 1916371, and NSF CAREER 1653404 for partially supporting this project. DP acknowledges NSF DMS 2210689 and NSF DMS 1916371 for partially supporting this project. YY acknowledges NSF DMS 2210717 for partially supporting this project.

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

# A. Appendix

## A.1. Proof of Lemma 3.1

*Proof.* Plugging the points $\mu_j^* = C_j n^{\frac{1-m}{m}}$, $1 \leq j \leq m$, $\mu_s^* = C_s n^{-1}$, $m+1 \leq s \leq d$, into the recursive formula

$$\mu_j^{(t+1)} = \prod_{s=1}^{j-1} G(\lambda_s, n\mu_s^{(t+1)}) \cdot \prod_{s=j+1}^{d} G(\lambda_s, n\mu_s^{(t)})$$

yields the system of equations

$$\prod_{j=1}^{m} C_j = \lambda^{m-1} \prod_{m+1 \leq s \leq d} G(\lambda_s, C_s), \quad C_s G(\lambda_s, C_s) = \lambda, \quad m+1 \leq s \leq d.$$

For each $m+1 \leq s \leq d$, $xG(\lambda_s, x)$ is monotonically increasing and $\lim_{x \to \infty} xG(\lambda_s, x) = \lambda_s > \lambda$, hence the equation $xG(\lambda_s, x) = \lambda$ has a unique solution. Any combination of $C_1, \ldots, C_m$ satisfying $\prod_{j=1}^{m} C_j = \lambda^{m-1} \prod_{m+1 \leq s \leq d} G(\lambda_s, C_s)$ will produce a fixed point for the system of equations 3.1. $\square$

## A.2. Proof of Lemma 3.2

The stability behavior of any nonlinear dynamical system near any hyperbolic fixed point can be determined by the eigenvalues of the Jacobian matrix. In order to determine the behavior CAVI system near its fixed points we need to first determine the Jacobian of the system and then the eigenvalues of the system at the fixed point.

We begin by defining some shorthand notation

$$R(\lambda, \beta) := \frac{\beta^{\lambda+1} \exp(-\beta)}{\Gamma(\lambda+1)\gamma(\lambda+1, \beta)} - \frac{\beta^{\lambda} \exp(-\beta)}{\Gamma(\lambda)\gamma(\lambda, \beta)}. \tag{A.1}$$

The derivative of $G$ with respect to $\beta$ is given by

$$\frac{d}{d\beta}G(\lambda, \beta) = G(\lambda, \beta)\left(-\frac{1}{\beta} + \frac{\beta^{\lambda} \exp(-\beta)}{\Gamma(\lambda+1)\gamma(\lambda+1, \beta)} - \frac{\beta^{\lambda-1} \exp(-\beta)}{\Gamma(\lambda)\gamma(\lambda, \beta)}\right) = \frac{G(\lambda, \beta)}{\beta}(-1 + R(\lambda, \beta)). \tag{A.2}$$

We are now equipped to begin computing the Jacobian of the system. We will compute expressions for these quantities recursively beginning with the partials of $\mu_1^{(t+1)}$. The partials of $\mu_1^{(t+1)}$ can be computed without recursion and are given by,

$$\partial_1 \mu_1^{(t+1)} = \frac{\partial}{\partial \mu_1^{(t)}} \prod_{s \neq 1} G(\lambda_s, n\mu_s^{(t)}) = 0$$

$$\partial_k \mu_1^{(t+1)} = \frac{\partial}{\partial \mu_k^{(t)}} \prod_{s \neq 1} G(\lambda_s, n\mu_s^{(t)}) = nG'(\lambda_k, n\mu_k^{(t)}) \prod_{s \neq 1, k}^{d} G(\lambda_s, n\mu_s^{(t)})$$

$$= \frac{\prod_{s \neq 1} G(\lambda_s, n\mu_s^{(t)})}{\mu_k^{(t)}}(-1 + R(\lambda_k, n\mu_k^{(t)}))$$

$$= \frac{\mu_1^{(t+1)}}{\mu_k^{(t)}}(-1 + R(\lambda_k, n\mu_k^{(t)})), \quad 2 \leq k \leq d.$$

For $j, k \in [d]$, the general derivatives of the system are given by the following equations. For $2 \leq k \leq j$,

$$\partial_k \mu_j^{(t+1)}(\mu) = \frac{\partial}{\partial \mu_k^{(t)}} \prod_{s=1}^{j-1} G(\lambda_s, n\mu_s^{(t+1)}) \cdot \prod_{s=j+1}^{d} G(\lambda_s, n\mu_s^{(t)})$$

$$= \sum_{1 \le \ell < j} nG'(\lambda_\ell, n\mu_\ell^{(t+1)}) \partial_k \mu_\ell^{(t+1)} \prod_{\substack{1 \le s \le j-1 \\ s \ne \ell}} G(\lambda_s, n\mu_s^{(t+1)}) \cdot \prod_{s=j+1}^{d} G(\lambda_s, n\mu_s^{(t)})$$

$$= \sum_{1 \le \ell < j} \frac{\prod_{s=1}^{j-1} G(\lambda_s, n\mu_s^{(t+1)}) \cdot \prod_{s=j+1}^{d} G(\lambda_s, n\mu_s^{(t)})}{\mu_\ell^{(t+1)}} (-1 + R(\lambda_\ell, n\mu_\ell^{(t+1)})) \partial_k \mu_\ell^{(t+1)}$$

$$= \sum_{1 \le \ell < j} \frac{\mu_j^{(t+1)}}{\mu_\ell^{(t+1)}} (-1 + R(\lambda_\ell, n\mu_\ell^{(t+1)})) \partial_k \mu_\ell^{(t+1)}.$$

and for $j + 1 \le k \le d$,

$$\partial_k \mu_j^{(t+1)} = \frac{\partial}{\partial \mu_k^{(t)}} \prod_{s=1}^{j-1} G(\lambda_s, n\mu_s^{(t+1)}) \cdot \prod_{s=j+1}^{d} G(\lambda_s, n\mu_s^{(t)})$$

$$= \sum_{1 \le \ell < j} nG'(\lambda_\ell, n\mu_\ell^{(t+1)}) \partial_k \mu_\ell^{(t+1)} \prod_{\substack{1 \le s \le j-1 \\ s \ne \ell}} G(\lambda_s, n\mu_s^{(t+1)}) \cdot \prod_{s=j+1}^{d} G(\lambda_s, n\mu_s)$$

$$+ nG'(\lambda_k, n\mu_k^{(t)}) \prod_{1 \le s \le j-1} G(\lambda_s, n\mu_s^{(t+1)}) \prod_{\substack{j+1 \le s \le d \\ s \ne \ell}} G(\lambda_s, n\mu_s^{(t)})$$

$$= \sum_{1 \le \ell < j} \frac{\prod_{s=1}^{j-1} G(\lambda_s, n\mu_s^{(t+1)}) \cdot \prod_{s=j+1}^{d} G(\lambda_s, n\mu_s^{(t)})}{\mu_\ell^{(t+1)}} \partial_k \mu_\ell^{(t+1)} (-1 + R(\lambda_\ell, n\mu_\ell^{(t+1)}))$$

$$+ \frac{\prod_{s=1}^{j-1} G(\lambda_s, n\mu_s^{(t+1)}) \cdot \prod_{s=j+1}^{d} G(\lambda_s, n\mu_s^{(t)})}{\mu_k^{(t)}} (-1 + R(\lambda_\ell, n\mu_k^{(t)}))$$

$$= \sum_{1 \le \ell < j} \frac{\mu_j^{(t+1)}}{\mu_\ell^{(t+1)}} (-1 + R(\lambda_\ell, n\mu_\ell^{(t+1)})) \partial_k \mu_\ell^{(t+1)} + \frac{\mu_j^{(t+1)}}{\mu_k^{(t)}} (-1 + R(\lambda_k, n\mu_k^{(t)})).$$

Due to the sequential nature of the CAVI system in Equation 3.1, the derivatives of the update function $\mu_j$ depend on the derivatives of the previous update functions $\mu_\ell$, $1 \le \ell < j$.

Now we derive the behavior for the other partial derivatives in an recursive fashion. Next we calculate the partials for $\mu_2^{(t+1)}$. First,

$$\partial_1 \mu_2^{(t+1)} = \sum_{1 \le \ell < 2} \frac{\mu_2^{(t+1)}}{\mu_\ell^{(t+1)}} (-1 + R(\lambda_\ell, n\mu_\ell^{(t+1)})) \partial_1 \mu_\ell^{(t+1)}$$

$$= \frac{\mu_2^{(t+1)}}{\mu_1^{(t+1)}} (-1 + R(\lambda_1, n\mu_1^{(t+1)})) \partial_1 \mu_1^{(t+1)} = 0.$$

Similarly,

$$\partial_1 \mu_3^{(t+1)} = \sum_{1 \le \ell < 3} \frac{\mu_3^{(t+1)}}{\mu_\ell^{(t+1)}} (-1 + R(\lambda_\ell, n\mu_\ell^{(t+1)})) \partial_1 \mu_\ell^{(t+1)}$$

$$= \frac{\mu_3^{(t+1)}}{\mu_1^{(t+1)}} (-1 + R(\lambda_1, n\mu_1^{(t+1)})) \partial_1 \mu_1^{(t+1)} + \frac{\mu_3^{(t+1)}}{\mu_2^{(t+1)}} (-1 + R(\lambda_2, n\mu_2^{(t+1)})) \partial_1 \mu_2^{(t+1)} = 0.$$

This gives us the recursive identity

$$\partial_1 \mu_k^{(t+1)} = \sum_{1 \le \ell < k} \frac{\mu_k^{(t+1)}}{\mu_\ell^{(t+1)}}(-1 + R(\lambda_\ell, n\mu_\ell^{(t+1)}))\partial_1 \mu_\ell^{(t+1)} = 0.$$

The second partial of $\mu_2^{(t+1)}$ is given by,

$$\partial_2 \mu_2^{(t+1)} = nG'(\lambda_1, n\mu_1^{(t+1)})\partial_2 \mu_1^{(t+1)} \prod_{\substack{1 \le s \le d \\ s \ne 1,2}} G(\lambda_s, n\mu_s^{(t)})$$

$$= \frac{G(\lambda_1, n\mu_1^{(t+1)})\prod_{3 \le s \le d} G(\lambda_s, n\mu_s^{(t)})}{\mu_1^{(t+1)}}(-1 + R(\lambda_1, n\mu_1^{(t+1)}))\partial_2 \mu_1^{(t+1)}$$

$$= \frac{\mu_2^{(t+1)}}{\mu_1^{(t+1)}}(-1 + R(\lambda_1, n\mu_1^{(t+1)}))\frac{\mu_1^{(t+1)}}{\mu_2^{(t)}}(-1 + R(\lambda_2, n\mu_2^{(t)}))$$

$$= (-1 + R(\lambda_1, n\mu_1^{(t+1)}))(-1 + R(\lambda_2, n\mu_2^{(t)})).$$

For the partial with the higher index than the update function $k > j$ we have to follow an iterative construction procedure starting with $\partial_k \mu_2^{(t+1)}$, $3 \le k \le d$, which expands as follows.

$$\partial_k \mu_2^{(t+1)} = \sum_{1 \le \ell < 2} \frac{\mu_2^{(t+1)}}{\mu_\ell^{(t+1)}}(-1 + R(\lambda_\ell, n\mu_\ell^{(t+1)}))\partial_k \mu_\ell^{(t+1)} + \frac{\mu_2^{(t+1)}}{\mu_k^{(t)}}(-1 + R(\lambda_k, n\mu_k^{(t)}))$$

$$= \frac{\mu_2^{(t+1)}}{\mu_1^{(t+1)}}(-1 + R(\lambda_1, n\mu_1^{(t+1)}))\partial_k \mu_1^{(t+1)} + \frac{\mu_2^{(t+1)}}{\mu_k^{(t)}}(-1 + R(\lambda_k, n\mu_k^{(t)}))$$

$$= \frac{\mu_2^{(t+1)}}{\mu_1^{(t+1)}}\frac{\mu_1^{(t+1)}}{\mu_k^{(t)}}(-1 + R(\lambda_1, n\mu_1^{(t+1)}))(-1 + R(\lambda_k, n\mu_k^{(t)})) + \frac{\mu_2^{(t+1)}}{\mu_k^{(t)}}(-1 + R(\lambda_k, n\mu_k^{(t)}))$$

$$= \frac{\mu_2^{(t+1)}}{\mu_k^{(t)}}R(\lambda_1, n\mu_1^{(t)})(-1 + R(\lambda_k, n\mu_k^{(t)})).$$

Next we calculate the partials for $\mu_3^{(t+1)}$. As above the first partial $\partial_1 \mu_3^{(t+1)} = 0$. The second partial is

$$\partial_2 \mu_3^{(t+1)} = \sum_{1 \le \ell < 3} \frac{\mu_3^{(t+1)}}{\mu_\ell^{(t+1)}}(-1 + R(\lambda_\ell, n\mu_\ell^{(t+1)}))\partial_2 \mu_\ell^{(t+1)}$$

$$= \frac{\mu_3^{(t+1)}}{\mu_1^{(t+1)}}(-1 + R(\lambda_1, n\mu_1^{(t+1)}))\partial_2 \mu_1^{(t+1)} + \frac{\mu_3^{(t+1)}}{\mu_2^{(t+1)}}(-1 + R(\lambda_2, n\mu_2^{(t+1)}))\partial_2 \mu_2^{(t+1)}$$

$$= \frac{\mu_3^{(t+1)}}{\mu_1^{(t+1)}}(-1 + R(\lambda_1, n\mu_1^{(t+1)}))\partial_2 \mu_1^{(t+1)}(\mu^{(t+1)})$$

$$+ \frac{\mu_3^{(t+1)}}{\mu_1^{(t+1)}}(-1 + R(\lambda_2, n\mu_2^{(t+1)}))(-1 + R(\lambda_1, n\mu_1^{(t+1)}))\partial_2 \mu_1^{(t+1)}$$

$$= \frac{\mu_3^{(t+1)}}{\mu_1^{(t+1)}}(-1 + R(\lambda_1, n\mu_1^{(t+1)}))R(\lambda_2, n\mu_2^{(t+1)})\partial_2 \mu_1^{(t+1)}$$

$$= \frac{\mu_3^{(t+1)}}{\mu_2^{(t+1)}}(-1 + R(\lambda_1, n\mu_1^{(t+1)}))(-1 + R(\lambda_2, n\mu_2^{(t+1)}))R(\lambda_2, n\mu_2^{(t+1)}).$$

The third partial is

$$\partial_3\mu_3^{(t+1)} = \sum_{1\le\ell<3}\frac{\mu_3^{(t+1)}}{\mu_\ell^{(t+1)}}(-1+R(\lambda_\ell,n\mu_\ell^{(t+1)}))\partial_3\mu_\ell^{(t+1)}$$

$$= \frac{\mu_3^{(t+1)}}{\mu_1^{(t+1)}}(-1+R(\lambda_1,n\mu_1^{(t+1)}))\partial_3\mu_1^{(t+1)} + \frac{\mu_3^{(t+1)}}{\mu_2^{(t+1)}}(-1+R(\lambda_2,n\mu_2^{(t+1)}))\partial_3\mu_2^{(t+1)}$$

$$= (-1+R(\lambda_1,n\mu_1^{(t+1)}))(-1+R(\lambda_3,n\mu_3^{(t+1)}))$$
$$+ R(\lambda_1,n\mu_1^{(t+1)})(-1+R(\lambda_2,n\mu_2^{(t+1)}))(-1+R(\lambda_3,n\mu_3^{(t+1)}))$$

$$= (-1+R(\lambda_3,n\mu_3^{(t+1)}))(-1+R(\lambda_1,n\mu_1^{(t+1)})R(\lambda_2,n\mu_2^{(t+1)})).$$

The $k$th partial, for $4\le k\le d$ is given by,

$$\partial_k\mu_3^{(t+1)} = \sum_{1\le\ell<3}\frac{\mu_3^{(t+1)}}{\mu_\ell^{(t+1)}}(-1+R(\lambda_\ell,n\mu_\ell^{(t+1)}))\partial_k\mu_\ell^{(t+1)} + \frac{\mu_3^{(t+1)}}{\mu_k^{(t)}}(-1+R(\lambda_k,n\mu_k^{(t)}))$$

$$= \frac{\mu_3^{(t+1)}}{\mu_1^{(t+1)}}(-1+R(\lambda_1,n\mu_1^{(t+1)}))\partial_k\mu_1^{(t+1)} + \frac{\mu_3^{(t+1)}}{\mu_2^{(t+1)}}(-1+R(\lambda_2,n\mu_2^{(t+1)}))\partial_k\mu_2^{(t+1)}$$

$$+ \frac{\mu_3^{(t+1)}}{\mu_k^{(t)}}(-1+R(\lambda_k,n\mu_k^{(t)}))$$

$$= \frac{\mu_3^{(t+1)}}{\mu_k^{(t)}}(-1+R(\lambda_1,n\mu_1^{(t+1)}))(-1+R(\lambda_k,n\mu_k^{(t)}))$$

$$+ \frac{\mu_3^{(t+1)}}{\mu_k^{(t)}}R(\lambda_1,n\mu_1^{(t+1)})(-1+R(\lambda_2,n\mu_2^{(t+1)}))(-1+R(\lambda_k,n\mu_k^{(t)}))$$

$$+ \frac{\mu_3^{(t+1)}}{\mu_k^{(t)}}(-1+R(\lambda_k,n\mu_k^{(t)}))$$

$$= \frac{\mu_3^{(t+1)}}{\mu_k^{(t)}}(-1+R(\lambda_k,n\mu_k^{(t)}))\prod_{1\le\ell\le2}R(\lambda_\ell,n\mu_\ell^{(t+1)}).$$

In general when $k>j$ we have,

$$\partial_k\mu_j^{(t+1)} = \frac{\mu_j^{(t+1)}}{\mu_k^{(t)}}\prod_{1\le\ell<j}R(\lambda_\ell,n\mu_\ell^{(t+1)})(-1+R(\lambda_k,n\mu_k^{(t)})).$$

For the partial with the same index as the update function we have,

$$\partial_k\mu_k^{(t+1)} = (-1+R(\lambda_k,n\mu_k^{(t+1)}))(-1+\prod_{1\le\ell<j}R(\lambda_\ell,n\mu_\ell^{(t+1)})).$$

Finally, when $k<j$ we have,

$$\partial_k\mu_j^{(t+1)} = \frac{\mu_j^{(t+1)}}{\mu_k^{(t+1)}}\prod_{k\le p<j}R(\lambda_p,n\mu_p^{(t+1)})(-1+\prod_{1\le\ell<k}R(\lambda_\ell,n\mu_\ell^{(t+1)}))(-1+R(\lambda_k,n\mu_k^{(t+1)})). \quad \square$$

## A.3. Singular Regression Example

Previous work by [39] and [31] on mean-field VI in singular models demonstrates that the leading coefficient of the asymptotic expansion of the ELBO is not the RLCT when the mean-field approximation is computed in the original coordinate system of the model. We will now provide a simpler example of a singular regression model for which the asymptotic behavior of the ELBO for the mean-field approximation in the original coordinate system does not behave like $-\lambda \log n + C$, but the ELBO for the mean-field approximation in the resolved coordinate system asymptotically behaves like $-\lambda \log n + C$.

We will consider the following singular regression model, Example 46 from [28], for $x, y \in \mathbb{R}$, and parameters $w = (a, b, c) \in [-1, 1]^3$,

$$p(x, y \mid a, b, c) = \frac{1}{2\sqrt{2\pi}} \exp\left\{-\frac{1}{2}(y - aS(bx) - cx)^2\right\} I_{[-1,1]^3}(x),$$

$$\phi(a, b, c) = 1/8,$$

where $S(x) = x^2 + x$. We will assume that the true data generating distribution is $q(x, y) = p(x, y \mid 0, 0, 0)$. The average log-density ratio for this setting is $K(a, b, c) = (ab + c)^2 + 3a^2b^4$. This example was chosen because it is one of the few examples for which the full resolution of singularities is explicitly known.

A real analytic function $f(w)$ is *normal crossing* if it has the form $f(w) = w^{2k}f_0(w)$, where k is a multi-index in $\mathbb{N}^d$ with at least one positive entry and $f_0(w)$ is a positive real-analytic function [26, Definition 4.3]. Notice that in the $(a, b, c)$-coordinate system the model $K(a, b, c)$ is not normal crossing, so the posterior is not in standard form in this coordinate system. Hironaka's theorem [25] guarantees that there exists a coordinate system in which $K$ is normal crossing. This coordinate system can be found through a finite sequence of blow-ups of the parameter space $W$ by a variety $V$, in which we replace the variety $V$ by a copy of the projective plane $\mathbb{P}^{n-1}$. See chapter 3 of [27] for additional technical details on the computation of blow-ups.

### A.3.1. Computing the Resolution

The first set in the resolution is to determine the singular structure of the hyper-surface $\{K(a, b, c) = 0\}$. Since this is a hyper-surface defined by a single polynomial, the singular points of $\{K = 0\}$ are the points of $W$ where the Jacobian of $K$ also vanishes. The partial derivatives of $K$ are

$$\frac{\partial K}{\partial a} = 2(ab + c)b + 6ab^4, \quad \frac{\partial K}{\partial b} = 2(ab + c)a + 12a^2b^3, \quad \frac{\partial K}{\partial c} = 2(ab + c).$$

Setting these equal to zero and solving the system shows that the singular locus is given by $V = \{a = 0, c = 0\} \cup \{b = 0, c = 0\}$.

We will now outline how the resolution is computed for this example. The first step is to compute the blow-up of $W$ by center $\{a = 0, c = 0\}$. This blow-up is covered by two local coordinate charts,

$$U_1 : a = a_1c_1, b = b_1, c = c_1, \quad U_2 : a = a_2, b = b_2, c = a_2c_2$$

In $U_1$, $K(a, b, c) = c_1^2\left[(a_1b_1 + 1)^2 + 3a_1^2b_1^4\right]$ is normal crossing, as the function $(a_1b_1 + 1)^2 + 3a_1^2b_1^4$ is smooth and positive on $U_1$. In $U_2$, $K(a, b, c) = a_2^2[(b_2 + c_2)^2 + 3b_2^4]$, which is not normal crossing in this coordinate system as the function $(b_2 + c_2)^2 + 3b_2^4$ is singular on the set $\{b_2 = 0, c_2 = 0\}$. In order to make $K$ normal crossing in the $U_2$ coordinate chart we will need to compute further blow-ups to resolve the singular part of $(b_2 + c_2)^2 + 3b_2^4$.

The next step is to blow-up coordinate system $U_2$ with center $\{b_1 = 0, c_1 = 0\}$. This blow-up is covered by two local coordinate charts,

$$U_3 : b_2 = b_3, c_2 = b_3c_3, \quad U_4 : b_2 = b_4c_4, c_2 = (1 - b_4)c_4.$$

The blow-up in $U_4$ is chosen cleverly to reduce $b_2 + c_2 = c_4$. In $U_3$, the original coordinates are

$$U_3 : a = a_2, b = b_3, c = a_2b_3c_3$$

and $K(a, b, c) = a_2^2 b_3^2[(1+c_3)^2 + 3b_3^4]$. In $U_3$, $K$ is not a normal crossing since $(1+c_3)^2 + 3b_3^2$ is singular at $\{c_3 = -1, b_3 = 0\}$. In the $U_4$ chart, the original coordinates are

$$U_4 : a = a_2, b = b_4 c_4, c = a_2(1 - b_4)c_4$$

and $K(a, b, c) = a_2^2 c_4^2(1 + 3b_4^2 c_4^2)$. In $U_4$, $K$ is normal crossing since $1 + 3b_4^2 c_4^2$ smooth and positive.

The final blow-up, $U_3$ with center $\{c_3 = -1, b_3 = 0\}$, begins by applying a change of notation $u = b_3$ and $v = c_3 + 1$. This makes

$$U_3 : a = a_2, b = u, c = a_2 u(v - 1)$$

with $K(a, b, c) = a_3^2 u^2[v^2 + 3u^2]$. The blow-up of $U_3$ at center $\{u = 0, v = 0\}$ is covered by two local charts

$$U_5 = \{u = b_5, v = b_5 c_5\}, \quad U_6 = \{u = b_6 c_6, v = c_6\}.$$

In $U_5$, the original $(a, b, c)$-coordinates are given by

$$U_5 : a = a_2 = a_5, b = b_5, c = a_5 b_5(b_5 c_5 - 1)$$

and $K(a, b, c) = a_5^2 b_5^4(c_5^2 + 3)$, which is normal crossing. In $U_6$, the original $(a, b, c)$-coordinates are given by

$$U_6 : a = a_2 = a_5, b = b_6 c_6, c = a_2 b_6 c_6(c_6 - 1)$$

and $K(a, b, c) = a_6^2 b_6^2 c_6^4[1 + 3b_6^2]$, which is normal crossing. This completes the resolution of the singularities for this example. Before proceeding we will relabel the local coordinate systems with the following permutation of notation to re-align our notation with [28]; For sub-indexes $j \in 4, 5, 6$ map $j \mapsto (j - 2)$.

The resolution map $w = g(u)$ is given locally, by

$$
\begin{aligned}
a &= a_1 c_1, & b &= b_1, & c &= c_1, \\
a &= a_2, & b &= b_2 c_2, & c &= a_2(1 - b_2)c_2, \\
a &= a_3, & b &= b_3, & c &= a_3 b_3(b_3 c_3 - 1), \\
a &= a_4, & b &= b_4, & c &= a_4 b_4 c_4(c_4 - 1).
\end{aligned}
$$

The image of the local coordinate system $U_j = \{(a_j, b_j, c_j)\}$ is denoted by $W_j$ with,

$$
\begin{aligned}
W_1 &= \{(a, b, c) : |a| \leq |c|\}, \\
W_2 &= \{(a, b, c) : |a| \geq |c|, |ab| \leq |ab + c|\}, \\
W_3 &= \{(a, b, c) : |a| \geq |c|, |ab + c| \leq |ab^2|\}, \\
W_4 &= \{(a, b, c) : |a| \geq |c|, |ab^2| \leq |ab + c| \leq |ab|\}.
\end{aligned}
$$

Notice that $W = \bigcup_j W_j$ and each local coordinate system $U_j = \{(a_j, b_j, c_j)\} = \overline{g^{-1}(W_j^o)}$, $j = 1, 2, 3, 4$, is the closure of the inverse image of the interior of $W_j$. On each local coordinate system the average log density ratio $K$ is normal crossing,

$$K(a, b, c) = c_1^2\left([a_1 b_1 + 1]^2/2 + a_1^2 b_1^4/6\right) = a_2^2 c_2^2(1/2 + b_2^2 c_2^2/6) = a_3^2 b_3^4(c_3^2/2 + 1/6) = a_4^2 b_4^2 c_4^2(1/2 + b_4^2/6)$$

and the determinant of the Jacobian matrix is given by

$$|g'(u)| = |c_1| = |a_2 c_2| = |a_3 b_3^2| = |a_4 b_4 c_4|^2.$$

The local RLCT $\lambda_j$ and multiplicity $m_j$ on $U_j$ are

$$(\lambda_1, m_1) = (1, 1), (\lambda_2, m_2) = (1, 2), (\lambda_3, m_3) = (3/4, 1), (\lambda_4, m_4) = 3/4, 1).$$

Therefore, the RLCT and multiplicity for this singular model are $\lambda = 3/4$ and $m = 1$.

### A.3.2. Computing the CAVI Equations

In the original $(a, b, c)$-coordinate system the CAVI approximation to the posterior arising from Equation 4.1 takes the form

$$\rho(a) \propto \exp\{\nu_{b,\,2}a^2/2 + \nu_{b,\,1}\nu_{c,\,1}a + \nu_{b,\,4}a^2/6\},$$
$$\rho(b) \propto \exp\{\nu_{a,\,2}b^2/2 + \nu_{a,\,1}\nu_{c,\,1}b + \nu_{a,\,2}b^4/6\},$$
$$\rho(c) \propto \exp\{c^2/2 + \nu_{a,\,1}\nu_{b,\,1}c\},$$

where $\nu_{s,\,k} = \int_{-1}^{1} s^k \rho(s)ds$, $s \in \{a, b, c\}$ and $k \in \{1, 2, 4\}$. A numerical investigation of the asymptotic behavior of the ELBO for this system is summarized in Table 3. Interestingly, we see that the CAVI ELBO $\Psi_n(\rho^*) = -\log n + C$. It appears that in the original $(a, b, c)$-coordinate system the ELBO of the CAVI fixed point is capturing the maximum $\lambda_j$ rather than the minimum one.

We now compare this to the behavior of the CAVI algorithm computed on each of the local resolution coordinates. In the $U_1$-coordinate system the CAVI approximation to the standard form of the posterior arising from Equation 4.1 takes the form

$$\rho(a_1) \propto \exp\{a_1^2 \nu_{b_1,2}\nu_{c_1,2}/2 + \nu_{b_1,1}\nu_{c_1,2}a_1 + \nu_{b_1,4}\nu_{c_1,2}a_1^2/6\},$$
$$\rho(b_1) \propto \exp\{\nu_{a_1,2}\nu_{c_1,2}b_1^2/2 + \nu_{a_1,1}\nu_{c_1,2}b_1 + \nu_{a_1,2}\nu_{c_1,2}b_1^4/6\},$$
$$\rho(c_1) \propto |c_1| \exp\{\nu_{a_1,2}\nu_{b_1,2}c_1^2/2 + \nu_{a_1,1}\nu_{b_1,1}c_1^2 + 1/2c_1^2 + \nu_{1,2}\nu_{b_1,4}c_1^2/6\},$$

where $\nu_{s,\,k} = \int_{-1}^{1} s^k \rho(s)ds$, $s \in \{a_1, b_1, c_1\}$ and $k \in \{1, 2, 4\}$.

In the $U_2$-coordinate system the CAVI approximation to the standard form of the posterior arising from Equation 4.1 takes the form

$$\rho(a_2) \propto |a_2| \exp\{\nu_{c_2,2}a_2^2/2 + \nu_{b_2,2}\nu_{c_2,4}a_2^2/6\},$$
$$\rho(b_2) \propto \exp\{\nu_{a_2,2}\nu_{c_2,4}b_2^2/6\},$$
$$\rho(c_2) \propto |c_2| \exp\{\nu_{a_2,2}c_2^2/2 + \nu_{a_2,2}\nu_{b_2,2}c^4/6\},$$

with $\nu_{s,k}$ following a similar notational convention as before. In the $U_3$-coordinate system the CAVI approximation to the standard form of the posterior arising from Equation 4.1 takes the form

$$\rho(a_3) \propto |a_3| \exp\{\nu_{b_3,4}a_3^2/6 + \nu_{b_3,4}\nu_{c_3,2}a_3^2/2\},$$
$$\rho(b_3) \propto |b_3|^2 \exp\{\nu_{a_3,2}b_3^4/6 + \nu_{a_3,2}\nu_{c_3,2}b_3^4/2\},$$
$$\rho(c_3) \propto \exp\{\nu_{a_3,2}\nu_{b_3,4}c_3^2/2\},$$

with $\nu_{s,k}$ following a similar notational convention as before. Finally, in the $U_4$-coordinate system the CAVI approximation to the standard form of the posterior arising from Equation 4.1 takes the form

$$\rho(a_4) \propto |a_4|^2 \exp\{\nu_{b_4,2}\nu_{c_4,2}a_4^2/2 + \nu_{b_4,4}\nu_{c_4,2}a_4^2/6)\},$$
$$\rho(b_4) \propto |b_4|^2 \exp\{\nu_{a_4,2}\nu_{c_4,2}b_4^2/2 + \nu_{a_4,2}\nu_{c_4,2}b_4^4/6)\},$$
$$\rho(c_4) \propto |c_4|^2 \exp\{\nu_{a_4,2}\nu_{b_4,2}c_4^2/2 + \nu_{a_4,2}\nu_{b_4,4}c_4^2/6)\},$$

with $\nu_{s,k}$ following a similar notational convention as before. A numerical investigation of the asymptotic behavior of the ELBO for this system is summarized in Table 3. As predicted by Theorem 3.1, we see that the CAVI ELBO on each local coordinate system recovers $\text{ELBO}(\rho_j^*) = -\lambda_j \log n + C$.

## A.4. Additional Simulations from Sec 4.1

Additional simulations for lower-dimensional models comparing the ELBO and the asymptotic expansion in Equation. 2.1 as approximations for the log marginal likelihood (logML) in for a $d = 2, 6, 12, 15$ dimensional models in standard form with RLCT $\lambda = 1/2$ and multiplicity $m = 1$. For each model, there is a wide regime of sample sizes $n$ in which the ELBO provides a better approximation to the logML than the asymptotic approximation $-\lambda \log n + (m-1) \log \log n$. For sample sizes beyond this regime the logML is better approximated by the asymptotic approximation than the ELBO. This is due to the constant bias term that arises in the ELBO. For each model, as $n \to \infty$, the approximation error for the ELBO converges to a constant that depends on the dimension of the model. Furthermore, these simulations suggest that the point at which the ELBO begins to diverge significantly from the logML depends on the dimension of the model; note that the point at which the asymptotic approximation error dips below the ELBO approximation error moves right as $d$ increases from $d = 4$ to $d = 20$.

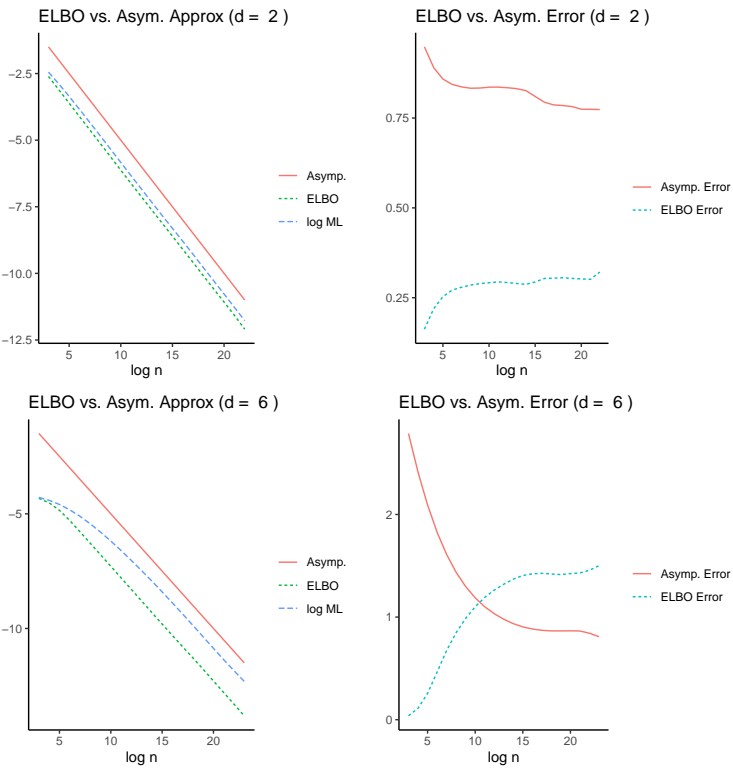

Figure 2: **Left**: A plot of the log marginal likelihood (logML), the ELBO, and Watanabe's asymptotic expansion for $d = 2, 6$, dimensional standard forms with RLCT $\lambda = 1/2$ and multiplicity $m = 1$. Surprisingly, in the non-asymptotic regime, the ELBO provides a better approximation of the log marginal likelihood than the asymptotic expansion. **Right**: A plot of the approximation errors between the logML and ELBO and logML and asymptotic approximation. As $n \to \infty$, both errors converge to constant values.

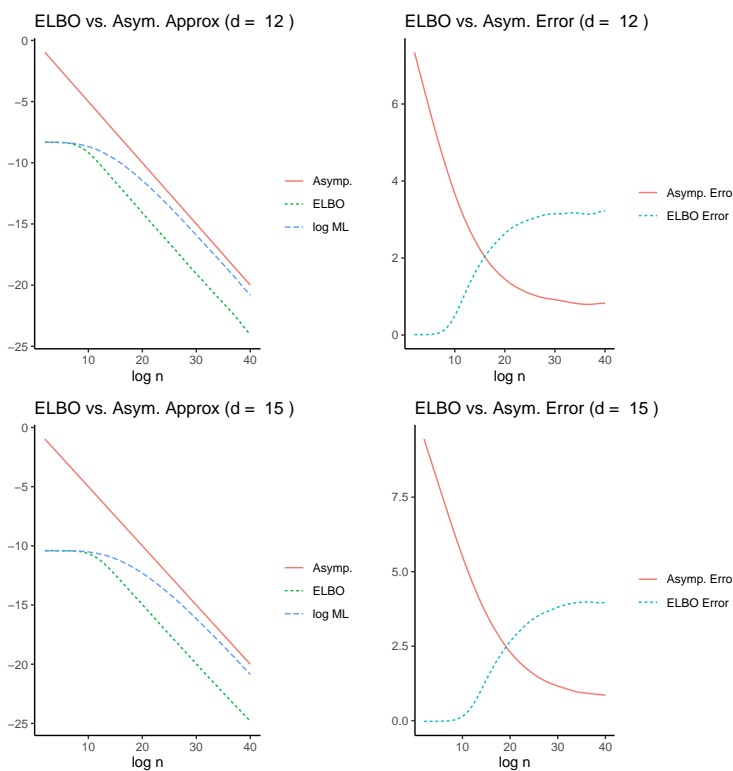

Figure 3: **Left**: A plot of the log marginal likelihood (logML), the ELBO, and Watanabe's asymptotic expansion for $d = 12, 15$ dimensional standard forms with RLCT $\lambda = 1/2$ and multiplicity $m = 1$. Surprisingly, in the non-asymptotic regime, the ELBO provides a better approximation of the log marginal likelihood than the asymptotic expansion. **Right**: A plot of the approximation errors between the logML and ELBO and logML and asymptotic approximation. As $n \to \infty$, both errors converge to constant values.

