# OpenReview forum: "Asymptotic Behavior of the Coordinate Ascent Variational Inference in Singular Models"
_CPAL.cc/2025/Proceedings_Track — CPAL 2025 (Proceedings Track) Poster_

### Official Review · Reviewer_Mmk5 · 2025-01-10
**Not qualified to review this**

**Rating:** 7
**Confidence:** 1

**Review:**

Thank you for considering me as a reviewer for this manuscript. After careful review, I feel that I may not be the most suitable reviewer for this particular work. My expertise does not sufficiently encompass the theoretical framework of singular models, specifically in the context of coordinate transformations and their role in variational inference.

The manuscript builds on several advanced concepts, such as real log-canonical thresholds (RLCT) and the resolution of singularities, which I am not well-versed in. Additionally, the work references key theoretical results and techniques, such as Bhattacharya’s asymptotic expansions and dynamical systems approaches, that I am less familiar with. As such, I am concerned that I might not be able to fully evaluate the motivation, technical depth, and implications of the proposed method.

In the best interest of the authors and the review process, I believe it would be more appropriate for a reviewer with stronger expertise in algebraic geometry, singular model theory, or variational inference methods to assess this work.

---

### Official Review · Reviewer_gogV · 2025-01-12
**Overall interesting work, presentation of experimental results could be improved**

**Rating:** 7
**Confidence:** 4

**Review:**

The submission studies the asymptotic behavior of coordinate ascent variational inference (CAVI) for mean-field approximations of singular models. The main theoretical results show that the ELBO computed in the standard form of the model retrieves the leading-order behavior of known asymptotic results. Experimental results support the theoretical results presented and highlight the advantage of using the standard form.

Strengths:
- The submission is well-written
- The studied problem is relevant to the community

Weaknesses:
- Theoretical results are presented in a dense way that may be hard to follow at times
- Presentation of experimental results could be improved

I will expand on these points below and I am looking forward to discussing with the authors!

---

1. Assumptions on $b(u)$. Could the authors expand on the conditions on $b(u)$ under which the presented results hold? It is stated on Line 144 that "we assume $b(u) = 1$. What happens when this assumption is not made? Does Theorem 3.1 still hold?

2. Experimental results.

General question on estimation of ELBO. What are the reference distributions $\mu$ used in the experiments to estimate their respective ELBOs?

- Sec 4.1.

What is the system being studied in this example? It may be helpful to include the standard form of the system. Also, am I understanding correctly that, in this experiment, the system is directly defined in standard form, so there is no need to compute the resolved coordinates?

Lines 253-256. It may be helpful to clarify and expand on this claim, which to a first-time reader may be confusing. Which "bias" term are the authors referring to? Why does it depend on the dimensionality of the system?

Lines 256-259. I am not sure I understand where these results are included in the main text of the submission. Figure 1 only includes results for $d = 20$.

Table 2. It may be helpful to reiterate what the coefficients in the regression are trying to approximate. Right now, it is hard to intuitively follow the claims made about the results. Also, I am not sure I understand how to verify that "the regression fails to correctly capture the multiplicity $m$ when $m > 1$" from the table.

- Sec. 4.2

Could the authors expand on how the resolved coordinates are computed in this experiment?

Table 3. Should the caption read $ELBO(q^*_j)$, missing index $j$?

---

**Minor comments**

- Line 27: Typo in "this distribution is computed an algorithm"
- Line 68: the acronym RLCT is used without definition
- Line 73: it may be helpful to reiterate what the resolution map is
- Footnote 1: Incomplete equation $L(w) \coloneqq$
- Line 93: It may be helpful to include $\mu$ in the definition of ELBO
- Line 143: what does the $K$ in $\gamma_K$ stand for?
- Paragraph 3.1.3: a table may help summarize the findings in this subsection, which at the moment are hard to follow
- Line 248: typo in "in for a $d = 20$"

---

### Meta-Review · Area_Chair_wF4H · 2025-02-03

**Recommendation:** Accept (Poster)
**Confidence:** 3

**Metareview:**

This paper studies the asymptotic behavior of coordinate ascent variational inference (CAVI) for mean-field approximations of singular models. The reviewer found the paper well written and the problem relevant to the community. My decision is acceptance as a poster.

---

### Decision · Program_Chairs · 2025-02-11

Accept (Poster)